# A Co-Culture-Based Multiparametric Imaging Technique to Dissect Local H_2_O_2_ Signals with Targeted HyPer7

**DOI:** 10.3390/bios11090338

**Published:** 2021-09-14

**Authors:** Melike Secilmis, Hamza Yusuf Altun, Johannes Pilic, Yusuf Ceyhun Erdogan, Zeynep Cokluk, Busra Nur Ata, Gulsah Sevimli, Asal Ghaffari Zaki, Esra Nur Yigit, Gürkan Öztürk, Roland Malli, Emrah Eroglu

**Affiliations:** 1Molecular Biology, Genetics and Bioengineering Program, Sabanci University, 34956 Istanbul, Turkey; melikesecilmis@sabanciuniv.edu (M.S.); yaltun@sabanciuniv.edu (H.Y.A.); yceyhun@sabanciuniv.edu (Y.C.E.); zeynepcokluk@sabanciuniv.edu (Z.C.); busraata@sabanciuniv.edu (B.N.A.); gulsahsevimli@sabanciuniv.edu (G.S.); asal@sabanciuniv.edu (A.G.Z.); 2Molecular Biology and Biochemistry, Gottfried Schatz Research Center, Medical University of Graz, 8036 Graz, Austria; johannes.pilic@medunigraz.at; 3Research Institute for Health Sciences and Technologies (SABITA), Istanbul Medipol University, 34810 Istanbul, Turkey; esra.ekmekcioglu.92@gmail.com (E.N.Y.); gozturk@medipol.edu.tr (G.Ö.); 4Department of Biotechnology, Gebze Technical University, 41400 Kocaeli, Turkey; 5Physiology Department, International School of Medicine, Istanbul Medipol University, 34810 Istanbul, Turkey; 6BioTechMed Graz, Mozartgasse 12/II, 8010 Graz, Austria; 7Nanotechnology Research and Application Center, Sabanci University, 34956 Istanbul, Turkey

**Keywords:** endothelial cell line, HyPer7, hydrogen peroxide, targeted biosensors, co-imaging, mitochondria, multiparametric imaging, multiplex live-cell imaging, stable cell lines

## Abstract

Multispectral live-cell imaging is an informative approach that permits detecting biological processes simultaneously in the spatial and temporal domain by exploiting spectrally distinct biosensors. However, the combination of fluorescent biosensors with distinct spectral properties such as different sensitivities, and dynamic ranges can undermine accurate co-imaging of the same analyte in different subcellular locales. We advanced a single-color multiparametric imaging method, which allows simultaneous detection of hydrogen peroxide (H_2_O_2_) in multiple cell locales (nucleus, cytosol, mitochondria) using the H_2_O_2_ biosensor HyPer7. Co-culturing of endothelial cells stably expressing differentially targeted HyPer7 biosensors paved the way for co-imaging compartmentalized H_2_O_2_ signals simultaneously in neighboring cells in a single experimental setup. We termed this approach COMPARE IT, which is an acronym for co-culture-based multiparametric imaging technique. Employing this approach, we detected lower H_2_O_2_ levels in mitochondria of endothelial cells compared to the cell nucleus and cytosol under basal conditions. Upon administering exogenous H_2_O_2_, the cytosolic and nuclear-targeted probes displayed similarly slow and moderate HyPer7 responses, whereas the mitochondria-targeted HyPer7 signal plateaued faster and reached higher amplitudes. Our results indicate striking differences in mitochondrial H_2_O_2_ accumulation of endothelial cells. Here, we present the method’s potential as a practicable and informative multiparametric live-cell imaging technique.

## 1. Introduction

A cell is a complex system of many structurally and functionally organized biochemical pathways interactively controlling information transfer and maintaining homeostasis. Simultaneously acquiring only a few of these parameters may significantly enhance the experiment’s performance, robustness, and read-out [1]. Recent advances in differently colored genetically encoded fluorescent biosensors have opened new doors for simultaneously studying signaling dynamics with precise spatial and temporal resolution in single cells, tissues, and whole organisms [2,3]. However, only a few studies—mostly experts in live-cell imaging—employ and further advance these powerful tools.

Multispectral imaging, for example, is a revealing approach that exploits spectrally distinct biosensors in a given cell for dissecting the relationship and causation of specific biochemical pathways [4,5]. Although informative, this approach is technically challenging and has its limitations. The visible spectrum of light will default to the actual number of combinations of biosensors, reasonably not more than three, on a conventional widefield microscope [6]. If imaged in single cells, some probes can be paired with hue-shifted biosensors for the same analyte [7]. However, genetically encoded biosensors are engineered protein-based constructs, so the probes’ sensitivity and dynamic range may be significantly affected in different cell compartments [8]. The proteome integrity, different redox status, macromolecular crowding, and differences in the pH in different locales are parameters that can further influence the probes’ activity [9,10,11]. Despite these difficulties, we have learned much about the relationship between various signaling pathways and metabolic activities by employing multispectral imaging [12,13,14,15].

Multiplex live-cell imaging approaches are agile techniques that allow simultaneous multicolor imaging experiments in live cells co-expressing different biosensors [14,16]. In a recent study, researchers developed a multiplex imaging approach employing post hoc spectral unmixing analysis by exploiting targeted blue and red fluorescent proteins as a barcoding system that permits recording up to 72 combinations of fluorescence resonance energy transfer (FRET)-based and green fluorescent protein (GFP)-based biosensors [17]. Another approach uses single-color probes for obsolete complex and laborious spectral unmixing and image processing. Detecting different biosensors of the same color permits synchronous detection of multiple parameters in aggregated protein clusters inside cells [18]. However, spatial and optical encoding of the various probes requires post hoc imaging experiments and image processing, such as immunostaining using specific epitopes fused to each biosensor. Additionally, localization of the probes to ultra-sites is less controllable and non-predictable, and overlapping regions may undermine the spatial detection of different parameters. In a recent advance, Werley et al. developed an elegant approach—termed MOSAIC—that exploits arrayed island of cells on a glass dish [19]. This informative tool allowed simultaneous recordings from 20 different sensors in cultured cells in a single experimental setup. Some advantages of this approach are that cells expressing different biosensors experience the same treatment such as same temperature, the same ligand administration, the same pH or imaging buffer effects, as well as avoiding well-to-well variations and minimal reagent and material consumption. However, the production of micro patterns on a microscope slide and microarray printing of lentiviral particles for cell-island boxes require sophisticated devices and expert knowledge to establish this technique. Additionally, this approach is challenging for single-cell resolution because the imaging setup requires a large field of view [19].

Such highly sophisticated imaging technologies may be revelatory but require laborious pre-experimental cell-preparation steps and post-imaging process algorithms. Because of these technical complexities, multiparametric imaging is less accessible to a broad audience of life scientists as a routine application. New user-friendly approaches, which retain the advantages for high-content cellular assays yet permit simple implementation on a standard imaging rig, would alleviate these obstacles. To overcome such technical barriers, we sought to devise a simple imaging technique that allows multiparametric read-outs yet disentangles from the complexities of highly sophisticated technologies. Thus, here we established a co-culture and co-imaging-based system with differentially targeted biosensors stably expressed in immortalized endothelial cells. We believe that such an approach identifies subtle characteristics of cellular responses if visualized in direct comparison in a single experiment. We dubbed this method COMPARE IT (co-culture-based multiparametric imaging technique). Our results demonstrate that this technique is easy to establish, transformable to any other mono- or multi-colored biosensor combination, robust, and reliable and may potentially turn into a standard live-cell imaging technique in the future.

## 2. Materials and Methods

### 2.1. Molecular Cloning and Lentivirus Generation

Differentially targeted HyPer7 constructs were subcloned into a 3rd-generation lentivirus shuttle vector pLenti-MP2 (Addgene #36097) using BamHI and XhoI restriction sites. Plasmids were amplified in chemically competent Stbl3 bacteria. HEK293T cells were used to generate lentivirus particles. At confluency of 80–90% cells were co-transfected with 3 µg psPAX2 (Addgene #12260), 3 µg pMD2.G (Addgene #12259), and 6 µg of the respective HyPer7 lentivirus shuttle vectors using PolyJet transfection reagent according to the manufacturer’s instructions (SignaGen Laboratories, Rockville, MD, USA). Twenty-four hours later, the transfection medium was replaced by fresh Dulbecco’s minimal essential medium (DMEM). After further 24 h and 48 h incubation, the cell culture medium containing virus particles was collected and filtered using a 0.45 µm low protein binding medium filter (T.P.P., Switzerland) and subsequently ultra-filtrated using a 100 kDa Amicon^®^ Ultra-15 Centrifugal Filter Unit (3000× *g*, 30 min, 4 °C), to concentrate the lentivirus particles. Filtrates were aliquoted and immediately used or snap-frozen in liquid nitrogen and stored at −80 °C.

### 2.2. Cell Culture and Stable Cell Line Generation

HEK293T cells (passage number <20) were grown in DMEM supplemented with 10% fetal bovine serum (FBS) (Pan-Biotech, Aidenbach, Germany), 100 µg/mL Penicillin (Pan-Biotech, Aidenbach, Germany), and 100 U/mL Streptomycin (Pan-Biotech, Aidenbach, Germany). EA.hy926 cell line (ATCC, CRL-2922, Manassas, VA, USA) was cultured in DMEM (ATCC, Manassas, VA, USA) supplemented with 10% FBS, 100 µg/mL Penicillin, 100 U/mL Streptomycin, 100 µg/mL Normocin (InvivoGen, San Diego, CA, USA), and 2% HAT ((Sodium Hypoxanthine (5 mM), Aminopterin (20 µM), and Thymidine (0.8 mM)) (ATCC, Manassas, VA, USA). Cells were maintained in a humidified CO_2_ incubator (5% CO_2_, 37 °C). EA.hy926 cells were seeded on a 6-well plate and allowed to attach before replacing the antibiotic-free transduction medium containing 10% FBS, 10 µg/mL Polybrene infection reagent (Sigma-Aldrich, St. Louis, MO, USA), and respective lentivirus particles encoding for differentially targeted HyPer7. Optimization of lentiviral transduction was achieved by using serial dilutions of the viral-particle-containing filtrates. Cells were maintained in the virus-containing medium for 48–72 h. After positive transduction, cells were further cultured for one week in fresh complete DMEM before fluorescence-assisted cell sorting (FACS). Top 30% of HyPer7 positive cells were selected by detecting the green fluorescence emission using an excitation wavelength of 488 nm laser (Filter type: BP 530/40 nm) on a B.D. Influx Cell Sorter. Sorted EA.hy926 cells consisting of a mixture of positively transduced cells were regularly maintained under cell culture conditions before imaging experiments. Stable EA.hy926 cells expressing differentially localized HyPer7 probes were mixed in a 1:1:1 ratio and seeded on a 30 mm glass coverslip (Glaswarenfabrik Karl Knecht Sondheim, Sondheim vor der Rhön, Germany) one day before the experiment. For transient transfection, EA.hy926 cells were seeded on 30 mm glass coverslips. Between 16 and 24 h later, 1 µg of each purified plasmid HyperRed and Hyper7, respectively, was co-transfected using 2.5 µL PolyJet transfection reagent according to manufacturer’s instructions.

### 2.3. Buffers and Solutions

All chemicals were purchased from NeoFroxx unless otherwise stated. To maintain cells outside of the incubation chamber, a cell storage buffer containing 2 mM CaCl_2_, 5 mM KCl, 138 mM NaCl, 1 mM MgCl_2_, 1 mM HEPES (Pan-Biotech, Aidenbach, Germany), 0.44 mM KH_2_PO_4_, 2.6 mM NaHCO_3_, 0.34 mM NaH_2_PO_4_, 10 mM D-Glucose, 0.1% MEM Vitamins (Pan-Biotech, Aidenbach, Germany), 0.2% essential amino acids (Pan-Biotech, Aidenbach, Germany), 100 µg/mL Penicillin (Pan-Biotech, Aidenbach, Germany), and 100 U/mL Streptomycin (Pan-Biotech, Aidenbach, Germany) was used. The pH was adjusted to 7.43 using 1 M NaOH. The cell storage buffer was sterilized using a 0.45 µm medium filter (Isolab, Germany). For live-cell imaging experiments, a HEPES-buffered solution was used consisting of 2 mM CaCl_2_, 5 mM KCl, 138 mM NaCl, 1 mM MgCl_2_, 10 mM HEPES, 10 mM D-Glucose, and pH was adjusted to 7.43 using 1 M NaOH.

### 2.4. Imaging Experiments

All live-cell imaging experiments were performed on an inverted widefield epi-fluorescence microscope Zeiss Axio Observer.Z1/7 (Carl Zeiss AG, Oberkochen, Germany) equipped with an LED light source Colibri 7 (423/44 nm, 469/38 nm, 555/30), Plan-Apochromat 20×/0.8 dry objective, Plan-Apochromat 40×/1.4 oil immersion objective, a monochrome CCD camera Axiocam 503, and a custom-made gravity-based perfusion system. HyPer7 signals were imaged by alternately exciting cells using a motorized dual-filter wheel equipped with beam splitters (FT455 (for HyPer low, F_420_) and FT495 (for HyPer high, F_490_)). Emissions were alternately collected using a bandpass filter (BP 525/50). HyPerRed emission was collected using the filter combinations FT570 (BS) and emission filter 605/70. During the experiments, control and data acquisition were executed using Zen Blue 3.1 Pro software (Carl Zeiss AG, Oberkochen, Germany). Administration and withdrawal of exogenous H_2_O_2_ were performed using a custom-made perfusion system connected to a metal perfusion chamber (NGFI, Graz, Austria).

### 2.5. Statistical Analysis

All acquired imaging data were analyzed using GraphPad Prism software version 5.04 (GraphPad Software, San Diego, CA, USA). All data were repeated at least in triplicate. The number of experiments is given as “N”, while and the number of cells imaged is indicated as “n”. For instance, 4/28 indicates N = 4 (quadruplicate) and n = 28 (number of cells imaged in this particular experiment). All statistical data are presented as ±SD in addition to the representative real-time traces shown as curves (if not indicated otherwise). For the statistical comparisons of multiple groups, one-way ANOVA analyses of variances with post-test Bonferroni (comparison of all pairs of columns) were performed. Statistical significances were considered significant and indicated with “*” or “#”.

## 3. Results

### 3.1. Development of a Co-Culture-Based Multiparametric Imaging Technique

We first co-transfected immortalized endothelial cells (EA.hy926) with differentially targeted HyPerRed [20] and HyPer7 [21]. Transient transfection of these hardly transfectable cells yielded only a few co-transfected cells in a visible field of view (Appendix A). Co-imaging of consecutive administration and withdrawal of low and high concentrations of exogenous H_2_O_2_ showed striking differences in the kinetics and amplitude between the two fluorescent biosensors, despite the same subcellular localizations (Appendix A). While the ultrasensitive HyPer7 responded to low extracellular H_2_O_2_ concentrations in the cytosol and mitochondria, HyPerRed failed to show any response in both compartments (Appendix A). The subsequent administration of high H_2_O_2_ concentration to cells evoked similar signals in both biosensors (Appendix A). These results underpin our assumption that different sensor characteristics of HyPer7 and HyPerRed complicate spectral co-imaging of intracellular H_2_O_2_.

To tackle this issue, we designed and tested COMPARE IT, a co-culture-based multiparametric imaging approach exploiting stable cell lines expressing differently targeted biosensors with identical spectral properties (Figure 1). In this approach, we first generated three different EA.hy926 cell lines that stably express nuclear-, cytosolic-, or mitochondria-targeted HyPer7 biosensors by employing second-generation lentiviral systems. To retain the endothelial cell line characteristics, we selected the upper 30% of positively transduced cells by FACS (Appendix A). This approach yielded three new cell lines stably expressing the respective HyPer7 biosensors in three different cell locales (Figure 1 and Appendix A). Mixing each of these stable cell lines in equal amounts resulted in homogeneously distributed cells expressing nuclear- (HyPer7-NLS), cytosolic- (HyPer7-NES), and mitochondria-targeted HyPer7 (mito-HyPer7), respectively (Figure 1 and Appendix A).

We next tested whether different cells in close vicinity can be imaged simultaneously under identical experimental conditions. Such an approach would allow multiparametric imaging of locale H_2_O_2_ signals in individual cells. To test this idea, we co-cultured all three cell types in a 1:1:1 ratio and incubated the mixed cell populations overnight before imaging experiments (Figure 1). Figure 2A,B and Appendix A show that all three cell types are localizable in a single random field of view of a mixed cell population. Even reducing the image resolution (i.e., higher binning and lower magnification; 20× objective) allowed optical resolution of the differentially targeted cell lines expressing either HyPer7-NLS, HyPer7-NES, or mito-HyPer7 (Figure 2A,B and Appendix A).

HyPer7 is a single fluorescent protein (FP)-based ratiometric H_2_O_2_ biosensor [21]. Oxidation causes an increase in fluorescence intensity when excited at 490 nm (F_490_) and a decrease in fluorescence upon excitation at 420 nm (F_420_). The fluorescence intensities in both channels in different cells expressing HyPer7-NES or mito-HyPer7 were comparable (Figure 2D). However, cells expressing HyPer7-NLS showed a striking heterogeneity in fluorescence intensities in both channels (F_490_ and F_420_), indicating variability in the nuclear HyPer7 expression levels (Figure 2C and Appendix A).

Notably, our approach also casts new light on the redox levels in subcellular locales of cells under normal conditions. While for all cellular compartments, a basal ratio value around 1 was detected (Figure 2C,D), the basal ratio values for the mitochondria-targeted HyPer7 displayed the lowest levels on average compared to the other compartments yet with the highest heterogeneity (Figure 2D). This observation points to cell-to-cell and organelle-to-organelle heterogeneities in H_2_O_2_ levels of endothelial cells under basal conditions. These are very small and difficult-to-recognize differences in live cells if visualized in separate experiments; thus, our results are critical and confirm that simultaneous imaging of co-cultured cells is an informative approach for direct comparison of multiple parameters.

### 3.2. The COMPARE IT Approach Unravels Faster and Higher H_2_O_2_ Signals in Mitochondria of Endothelial Cells

Another promising finding we revealed with this approach is that administration of low extracellular H_2_O_2_ levels evoked an apparent increase of HyPer7 signal in all cell compartments. However, the amplitude and rate of fluorescence change of HyPer7 were strikingly different among the nucleus, cytosol, and mitochondria (Figure 3A,B).

Surprisingly, HyPer7 signals in response to extracellular H_2_O_2_ were lowest in the nucleus, followed by the cytosol, and highest within the mitochondria (Figure 3A,B). These differences in local H_2_O_2_ signals were more pronounced in response to low extracellular H_2_O_2_ concentrations, which increased the mitochondrial HyPer7 ratio twice as fast as the nucleus and cytosol (Figure 3A,B). We further observed that after washout of H_2_O_2_, the HyPer7 signals within the cell nucleus and cytosol almost fully recovered to basal levels within 5 min (Figure 3A,B). Under the same experimental conditions, the mitochondrial HyPer7 signal decreased with similar kinetics but remained above the basal ratio before the second administration of exogenous H_2_O_2_ (Figure 3A,B). Kinetic analyses unveiled that the rate of the HyPer7 ratio over time was also higher for the mitochondria-targeted HyPer7 compared to nuclear and cytosolic-targeted probes (Figure 3C–E). Administration of low extracellular H_2_O_2_ caused an instant mito-HyPer7 ratio increase, which plateaued within approximately 18 s. In contrast, the nuclear- and cytosolic-targeted HyPer7 remained almost unaffected within the first 18 s of exogenous H_2_O_2_ addition (Figure 3C,D). As expected, the rate of HyPer7 signals in all cell compartments instantly plateaued upon subsequent provision of high H_2_O_2_ concentration, while mito-HyPer7 again showed significantly faster kinetics in response to this cell treatment (Figure 3C,E). These findings summarize the benefit of the co-culture-based multiparametric imaging approach again.

## 4. Discussion

This study devised an easy-to-implement yet informative co-culture-based live-cell imaging technique that permits the acquisition of multiple parameters in a single experimental setup. We used lentiviral transduction methods to generate three new endothelial cell lines expressing differentially targeted HyPer7 (nucleus, cytosol, and mitochondria), which we utilized for multiparametric imaging experiments on a conventional widefield microscope. A gravity-driven perfusion system allowed the provision and withdrawal of different imaging buffers during the experiment [15]. Because this technique permits synchronous visualization and direct comparison of various cells, we dubbed this technique COMPARE IT. This method enabled us to quantitatively record and compare spatio-temporal patterns of transient and local intracellular H_2_O_2_ signals among neighboring cells under identical conditions.

Targeting HyPer7 to distinct cellular locales allowed spatial resolution of co-cultured cells, of which we took advantage and applied spectrally identical biosensors. Notably, this is a critical yet straightforward add-on obviating the need for complicated spectral unmixing or post hoc image processing. In contrast, multispectral imaging techniques employ distinct fluorescent biosensors (genetically encoded or chemical-based) [22,23], usually in the same cells. Although very informative, technical challenges (i.e., combining suitable filter settings and establishing complicated spectral unmixing algorithms) undermine implementing these methods.

We and others have shown in previous studies that different fluorophores significantly impact the sensitivity, dynamic range, and kinetic properties of re-engineered biosensors despite using the same sensor domains [7,20]. Thus, correct data interpretation requires additional control experiments. Here, we demonstrated the natural limitations of such multispectral imaging approaches by combining the novel HyPer7 [21] and the well-established HyPerRed [20]. Either in the same cell compartment or differentially targeted to the cytosol and mitochondria, respectively, any combination of co-expressing both probes yielded different signals, owing to the probe’s different sensitivities for H_2_O_2_ [20,21]. Both HyPerRed and HyPer7 consist of the sensor element OxyR, which are parts of a transcription factor sensitive for H_2_O_2_, derived from *E.coli* and *N. meningitidis*, respectively. Although both domains are derived from different species, they are a similar in structure but significantly different their sensitivity towards H_2_O_2_ [21,24]. In this context, it is obvious that spectral imaging has some limitations and should be used with caution to prevent data misinterpretation.

We sought to solve this issue by exploiting spectrally identical biosensors. As a result, we generated stable endothelial cell lines expressing differentially targeted HyPer7. Following cell sorting, we adopted an unconventional way and mixed the three cell lines with the aim to co-image all differentially targeted biosensors in a single field of a microscopic view allowing for synchronous imaging. Indeed, our approach proved to be suitable. Mixing equal amounts of cells yielded a homogenous distribution of cells expressing differently targeted HyPer7 biosensors that remained constant over time (i.e., approximately 24 h). We also observed that all three cell populations had similar proliferation properties. However, we always co-seeded cells 24 h before the experiment. Thus, one concern about this approach that is still elusive whether a 1:1:1 ratio remains constant over a longer time of passaging, freezing, and thawing cycles. Co-culturing cells with fluorescent reporters is a favored method employed in many studies [22,23,25]; however, to our knowledge, such an approach as presented in this study has not been shown elsewhere as a multiparametric imaging technique.

Having established COMPARE IT, we next sought to visualize H_2_O_2_ uptake and its diffusion in endothelial cells into cell compartments in direct comparison among neighboring cells. Former studies emphasize that the stable reactive oxygen species (ROS) H_2_O_2_ can readily diffuse into cells [26,27] to modulate diverse redox-sensitive proteins and cell functions, thus serving as an essential signaling molecule [28]. Several studies suggest that different aquaporin isoforms facilitate H_2_O_2_ diffusion across the plasma membrane and intracellular membranes of cellular organelles [29,30,31]. Our approach verifies such observations that even low extracellular concentrations of H_2_O_2_ can readily enter the cell membrane confirmed with the ultrasensitive HyPer7. Much to our surprise, we made an unexpected observation as the HyPer7 signals in the mitochondria appeared significantly earlier compared to the HyPer7 signals in the nucleus or cytosol. One would expect that with the sequential diffusion across a cell, the cytosolic-targeted probe would signal first. However, in response to both low and high concentrations of exogenously applied H_2_O_2_, HyPer7 ratio increased in the mitochondria first. Our approach raised an important question: how is it possible that exogenous H_2_O_2_ causes a clear temporally different HyPer7 signal in the mitochondria? It is generally accepted that different subcellular compartments harbor different antioxidant systems and redox capacities [32,33]. Given such variations in the redox balance and the permeability of membranes for H_2_O_2_ and ROS, buffering capacity might explain our observations to a certain degree. Further experiments are essential to address the diffusion of extracellular H_2_O_2_ across the plasma membrane and the two mitochondrial membranes into the organelles’ matrix. Additionally, expanding H_2_O_2_ diffusion studies among cell organelles [34] generated with chemogenetic tools might shed light on this issue [35,36]. However, it might be a challenge if the redox environment within mitochondria, harboring the electron transport chain complexes, alters the HyPer7 properties, rendering the fluorescent biosensor to an even more sensitive and rapid H_2_O_2_ indicator. In this study, we did not further address these interesting biological questions.

Here, we have developed a co-culture-based live-cell imaging technique using stable cell lines expressing differentially targeted biosensors that permits co-imaging of several parameters in a single experimental setup. This study utilized the novel HyPer7 biosensors to demonstrate the method’s potential as a practicable multiparametric imaging technique. Any conventional widefield microscope is suitable for establishing this simple approach using mono- or multicolor-based biosensors without technical or instrumental challenges.

## Figures and Tables

**Figure 1 biosensors-11-00338-f001:**
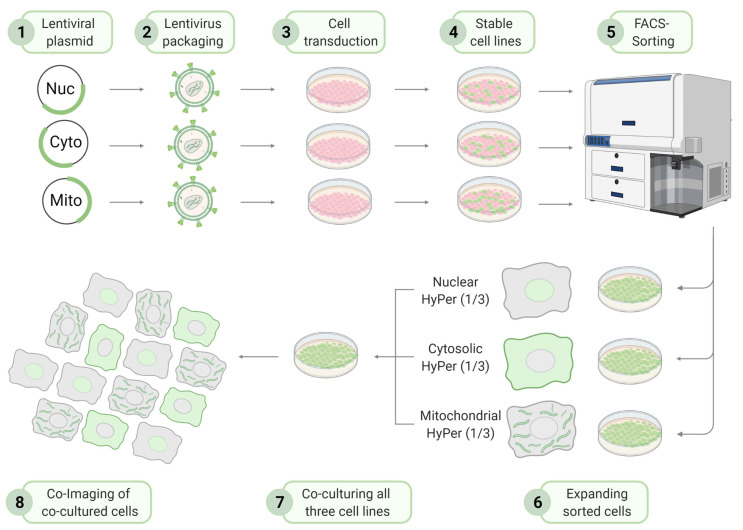
Schematic overview and workflow of the COMPARE IT approach. Step 1: The open reading frame of genes of interest (nuclear-, cytosolic-, and mitochondria-targeted HyPer7) are subcloned into a shuttle vector. Step 2: Lentiviral particles are generated and purified. Step 3: Native endothelial cells (EA.hy926) at low passage numbers are transduced with the respective purified lentivirus. Step 4: 48 to 72 h later, cells are further cultured and passaged for one week to allow cells to regenerate before being separated for positively transduced cells using FACS (Step 5). Step 6: Each cell line consisting of a mixture of positively transduced clones is further cultured for two more weeks or frozen for long-term storage. Step 7: For co-culture imaging, each cell line (nuclear-, cytosolic-, and mitochondria-targeted HyPer7) is seeded in a 1:1:1 ratio 24 h before imaging experiments to reach a confluency of 90% on the day of imaging. Step 8: Co-imaging of all three cell lines simultaneously using conventional fluorescence microscopy.

**Figure 2 biosensors-11-00338-f002:**
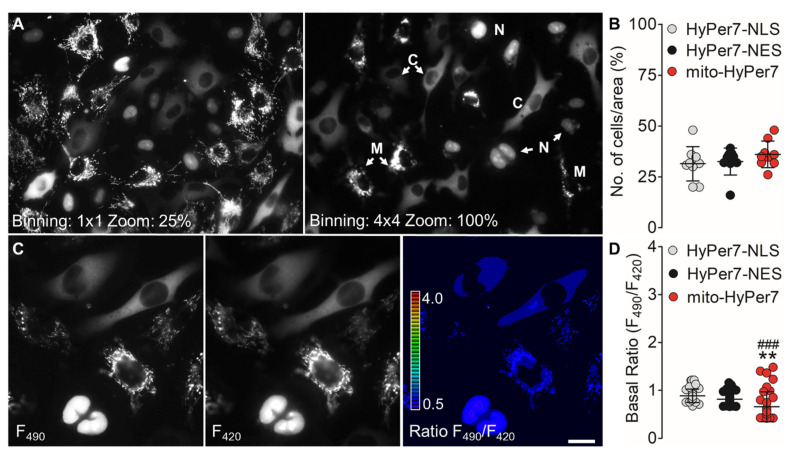
Characterization and analysis of co-cultured cell lines. (**A**) Micrographs show the distribution of co-cultured EA.hy926 cells expressing nuclear-, cytosolic-, and mitochondria-targeted HyPer7. The representative widefield images were captured using high-resolution settings (binning of 1 × 1, left image) and low-resolution settings (binning 4 × 4, right image). Subcellular localization is indicated with N for nucleus, C for cytosol, and M for mitochondria. Both images were taken with a 40× oil objective. (**B**) Scatter dot plot represents the statistical analysis of the distribution of differently targeted cells in percent in a random field of view. Nuclear-targeted HyPer7 (grey dots, n = 9/76), cytosolic-targeted HyPer7 (black dots, n = 9/68), and mitochondria-targeted HyPer7 (red dots, n = 9/76). (**C**) Representative widefield images show EA.hy926 cells expressing differentially targeted HyPer7 in the HyPer high channel (left image, Ex: 490 nm, Em: 520 nm) and HyPer low channel (middle image, Ex: 420 nm, Em: 520 nm). The right image shows the resulting ratio images (HyPer high channel divided by the HyPer low channel). Scale bar represents 20 µm. (**D**) Scatter dot plot shows the statistical analysis of the basal ratio levels of cells expressing the nuclear-targeted HyPer7 (grey dots, n = 38), cytosolic-targeted HyPer7 (black dots, n = 35), and mitochondria-targeted HyPer7 (red dots, n = 36). One-way ANOVA and Bonferroni’s multiple comparison post-test were applied to compare all columns with each other. *p* < 0.0001 *p*-value summary (Basal ratios: Cyto vs. mito **, Mito vs. Nuc ###, Cyto vs. Nuc n.s.). All values are given as ±SD.

**Figure 3 biosensors-11-00338-f003:**
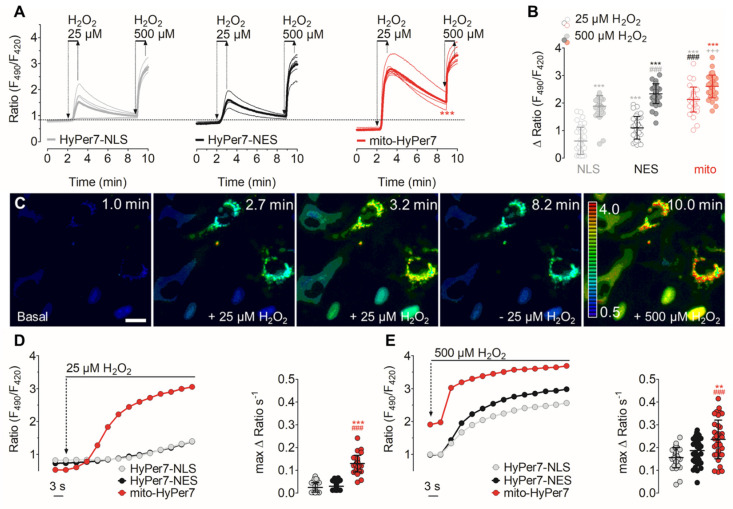
Multiparametric live-cell imaging of H_2_O_2_ in co-cultured cell lines. (**A**) Real-time traces of HyPer7 signals ((nuclear-targeted (grey curves, n = 6), cytosolic-targeted (black curves, n = 6), and mitochondria-targeted (red curves, n = 7)) in response to administration and withdrawal of 25 µM and 500 µM exogenous H_2_O_2_. All experiments are derived from a single field of view. (**B**) Scatter dot plot shows statistical analysis from panel (**A**); maximum HyPer7 ratio amplitudes in response to 25 µM (nuclear-targeted HyPer7, clear white dots, n = 5/36; cytosolic-targeted HyPer7, clear grey dots, n = 5/38; mitochondria-targeted HyPer7, clear red dots, n = 5/35) and 500 µM (nuclear-targeted HyPer7, full light grey dots, n = 5/36; cytosolic-targeted HyPer7, full dark grey dots, n = 5/38; mitochondria-targeted HyPer7, full red dots, n = 5/35). (**C**) Representative widefield ratio images (generated by dividing F_490_ images by F_420_ images) under basal conditions (very left image) in response to 25 µM H_2_O_2_ at different times points as indicated (second and third image), upon withdrawal of 25 µM H_2_O_2_ (fourth image) and in response to 500 µM H_2_O_2_ (right image). (**D**,**E**) Left panels show the overlay of real-time traces of HyPer7 signals in response to 25 µM (panel **D**) or 500 µM H_2_O_2_ (panel **E**) (nuclear HyPer7, grey curve; cytosolic HyPer7, black curve; mitochondrial HyPer7, red curve). Right panels show scatter dot plots and statistical analysis of the maximum ΔRatio per second of HyPer7 in the respective cellular locale (nucleus, grey dots, (n = 5/36); cytosol, black dots, (n = 5/38), and mitochondria, red dots, n = 5/35). One-way ANOVA and Bonferroni’s multiple comparison post-test were applied to compare all columns with each other. *p* < 0.0001 *p*-value summary (25 µM: Cyto vs. mito ***, Mito vs. Nuc ###, Cyto vs. Nuc n.s.); (500 µM: Cyto vs. mito **, Mito vs. Nuc ###, Cyto vs. Nuc n.s.). All values are given as ±SD.

## Data Availability

Data are available upon reasonable request.

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
