# Peer review of "A Co-Culture-Based Multiparametric Imaging Technique to Dissect Local H2O2 Signals with Targeted HyPer7"

_biosensors, 2021, doi:10.3390/bios11090338_

Round 1
Reviewer 1 Report
Secilmis et al. discuss a co-culture based imaging method to reveal details of molecular concentration in various cellular locations using a simple imaging platform. For this purpose, they target H2O2 as the molecule of concern and HyPer7 as the molecular biosensor. I found the manuscript of interest and well discussed.
I have couple minor comments indicated below:
- Abstract, line 19: “…the combination of different fluorescent biosensors with specific characteristics can undermine accurate co-imaging of the same analyte…” The phrase specific characteristics is vague here and provides no information. Consider rewording.
- Line 341: The authors state that method presented such as by them has been “less explored” as a multiparametric imaging technique – which indicates that this is not the first time this has been shown. Can they be more specific? What was the limitation of previous studies exploring this method as an imaging technique?
- Can the authors comment on the trend seen in Figure S1 using HyPerRed. While HyPerRed seems to closely match the ratiometric signal provided by HyPer7 at high concentrations, there is no response at all at 25 µM H2O2. Can the authors indicate the mechanism behind the threshold concentration for HyPerRed?
Author Response
Response to Reviewer 1 Comments
Point 1: Secilmis et al. discuss a co-culture based imaging method to reveal details of molecular concentration in various cellular locations using a simple imaging platform. For this purpose, they target H2O2 as the molecule of concern and HyPer7 as the molecular biosensor. I found the manuscript of interest and well discussed. I have couple minor comments indicated below:
Response 1: We appreciate reviewer #1’s positive feedback and suggestions to further improve our manuscript. All concerns have been edited in the revised version of the manuscript as indicated below. Respective changes are highlighted in the manuscript text.
Point 2: Abstract, line 19: “…the combination of different fluorescent biosensors with specific characteristics can undermine accurate co-imaging of the same analyte…” The phrase specific characteristics is vague here and provides no information. Consider rewording.
Response 2: We agree and have rephrased this sentence as follows and believe that this statement is now more accurate: “However, the combination of fluorescent biosensors with distinct spectral properties such as different sensitivities, and dynamic ranges can undermine accurate co-imaging of the same ana-lyte in different subcellular locales”. Please see respective changes on page 1, (lines 19-21).
Point 3: Line 341: The authors state that method presented such as by them has been “less explored” as a multiparametric imaging technique – which indicates that this is not the first time this has been shown. Can they be more specific? What was the limitation of previous studies exploring this method as an imaging technique?
Response 3: Indeed, we did not find any previous studies showing comparable approaches other than the ones presented in detail in the introduction part. Therefore, we rephrased the sentence as followed and believe that this points our statement more accurately than before. ;”…. however, to our knowledge, such an approach as presented in this study has not been shown elsewhere as a multiparametric imaging technique.” Please find respective changes on page 9 (lines 393 – 395)
Point 4: Can the authors comment on the trend seen in Figure S1 using HyPerRed. While HyPerRed seems to closely match the ratiometric signal provided by HyPer7 at high concentrations, there is no response at all at 25 µM H2O2. Can the authors indicate the mechanism behind the threshold concentration for HyPerRed?
Response 4: We thank reviewer 1 for this important note and clarified this issue in our manuscript text at page 9, lines 330-333 and added additional references demonstrating the differences in sensitivities of OxyR domains derived from various organisms.
New Ref 36: Sainsbury S, Ren J, Nettleship JE, Saunders NJ, Stuart DI, Owens RJ. The structure of a reduced form of OxyR from Neisseria meningitidis. BMC Struct Biol. 2010 May 17;10:10.
Reviewer 2 Report
In the submitted article, the authors constructed stable cell lines expressing nuclear, mitochondria and cytosol targeted H2O2 HyPer7 protein sensors. They co-cultured the cell lines expressing different localized sensors and used this mixed cells to detect differential stress in one culture dish. The work was well present and written. I found this work suited for this journal but the authors need to address the following concerns before it was published.
First, the sorting results should be included to present the stable line transduction efficiency.
Second, what is the rationale to detect these signals in one sample? Why cannot one just used one sensor at a time? The authors may need to demonstrate this.
Third, the colocalization of nuclear with DAPI and Mitochrodia with mito sensor probes should be used to validate the results.
Author Response
Response to Reviewer 2 Comments
Point 1: In the submitted article, the authors constructed stable cell lines expressing nuclear, mitochondria and cytosol targeted H2O2 HyPer7 protein sensors. They co-cultured the cell lines expressing different localized sensors and used this mixed cells to detect differential stress in one culture dish. The work was well present and written. I found this work suited for this journal but the authors need to address the following concerns before it was published.
Response 1: We appreciate reviewer #2’s positive feedback. The concerns of reviewer #2 have been edited in the revised version of the manuscript, and a new supplementary figure has been provided as indicated below.
Point 2: First, the sorting results should be included to present the stable line transduction efficiency.
Response 2: We have provided additional supplementary data; please see new supplementary figure 2, in which we now demonstrate the exquisite transduction efficiency of the hardly transfectable EA.hy926 cells.
Point 3: Second, what is the rationale to detect these signals in one sample? Why cannot one just used one sensor at a time? The authors may need to demonstrate this.
Response 3: The value of simultaneously imaging multiple parameters is that a single experiment yields different read-outs that can be collected in a single run for which in our case at least triplicate experiments would be necessary. Notably, this is our core message and we believe that we have extensively introduced, demonstrated, and discussed this issue throughout the entire manuscript such as in main figures 1, 2, and 3. Moreover, we explicitly point to this issue in the main text in lines 90-91, 301-303, 307-309, and 374-378.
Point 4: Third, the colocalization of nuclear with DAPI and Mitochrodia with mito sensor probes should be used to validate the results.
Response 4: We believe that the high-resolution images we have provided in Figure 2, Figure 3, supplementary figures 2 and 3 clearly demonstrate the correct localization of the targeted HyPer7 probes in the cell lines that stably express the differently localized H2O2 biosensors. Notably, the correct targeting of HyPer7 to mitochondria, cytosol and the nucleus respectively has been validated already in the original paper, which introduced HyPer7 for the first time; please see Pak, V.V.; Ezeriņa, D.; Lyublinskaya, O.G.; Pedre, B.; Tyurin-Kuzmin, P.A.; Mishina, N.M.; Thauvin, M.; Young, D.; Wahni, K.; Martínez Gache, S.A.; et al. Ultrasensitive Genetically Encoded Indicator for Hydrogen Peroxide Identifies Roles for the Oxidant in Cell Migration and Mitochondrial Function. Cell Metabolism 2020, 31, 642-653.e6.
Thus, we think that further validation experiments regarding the correct subcellular localization of targeted HyPer7 constructs are not necessary. We very much hope that reviewer 2 agrees with our assumption.
Reviewer 3 Report
Secilmis. and colleague created cytosol, mitochondria, and nucleus-targeted Hper7 expressing cells and imaged these cells with a co-culture method. I understand the manuscript, I do not have much concerns about it.
Author Response
Response to Reviewer 3 Comments
Point 1: Secilmis. and colleague created cytosol, mitochondria, and nucleus-targeted Hper7 expressing cells and imaged these cells with a co-culture method. I understand the manuscript, I do not have much concerns about it.
Response 1: We appreciate the positive feedback of reviewer #3 very much.